# A Novel Leaf-Derived Trapping Material Is More Effective at Capturing Common Bed Bugs (Hemiptera: Cimicidae) than Selected Commercial Monitoring Devices

**DOI:** 10.3390/insects16040362

**Published:** 2025-04-01

**Authors:** Jorge Bustamante, Patrick Liu, Kathleen Campbell, Andrew M. Sutherland, Dong-Hwan Choe, Catherine Loudon

**Affiliations:** 1Department of Entomology, University of California, Riverside, CA 92521, USA; kathleen.campbell@ucr.edu (K.C.); dchoe003@ucr.edu (D.-H.C.); 2Department of Ecology and Evolutionary Biology, University of California, Irvine, CA 92697, USA; liuph@uci.edu (P.L.); cloudon@uci.edu (C.L.); 3Cooperative Extension, Division of Agriculture and Natural Resources, University of California, Hayward, CA 94544, USA; amsutherland@ucanr.edu

**Keywords:** bed bug control, monitoring devices, sticky trap, pitfall trap

## Abstract

The early detection and monitoring of bed bug infestations is an important component of urban integrated pest management programs. This study evaluates a new bed bug trapping material derived from plant leaves, which uses tiny hair-like structures to trap bed bugs as they move across its surface. We compared the efficacy of this new trapping material to several commonly used commercial monitoring devices: a pitfall trap and three sticky traps. Our findings indicate that the leaf-derived material catches more bed bugs, especially in the juvenile stages, than the commonly used devices. This new trapping material might be useful in developing new bed bug detection and monitoring strategies.

## 1. Introduction

Bed bugs (*Cimex lectularius* L.) are small blood-feeding insects [1] that have resurged globally as important commensal pests in recent decades [2,3]. These pests feed when their vertebrate hosts are sleeping or resting and, at all other times, are known to hide in cracks and crevices near their hosts’ resting places, exhibiting positive thigmotaxis [1]. The resurgence of bed bugs has necessitated increased efforts in their management, early detection, and monitoring [3,4,5,6]. Pest management professionals often consider bed bugs the most challenging pest to control [7]. Moreover, bed bugs are often misidentified [8,9] or remain unnoticed for an extended period before an infestation becomes evident [10]. For decades, insecticides have been the most common pest control method for bed bugs [11,12]. However, many field populations of bed bugs have evolved insecticide resistance to pyrethroids and neonicotinoids [13].

Bed bug control methods include steam treatment, physical removal, mass trapping, and the encasement of mattresses, which are most effective when infestation levels are low [14,15]. Heat treatment has been established as an effective method for controlling bed bugs [16]. Desiccant dusts have also been helpful in bed bug control by disrupting their wax layer, causing them to lose water [17,18]. The early detection of bed bug infestations is crucial in effectively controlling bed bugs [5]. Furthermore, there is a growing need for enhanced monitoring and detection methods to effectively aid integrated pest management (IPM) plans [3,6].

Many integrated pest management plans for bed bugs currently incorporate passive monitoring and detection methods, such as pitfall and sticky traps [19]. Pitfall traps for bed bugs, often called “interceptors”, have proven highly effective at capturing bed bugs, even at low level infestations [8,20,21,22,23]. In contrast, sticky traps often perform poorly [23,24,25], possibly due to bed bugs’ aversion to the adhesive after making contact, inadequate adhesive strength, or environmental factors like dust accumulation that diminish trap efficacy over time [8]. Pest control operators may still utilize sticky traps since they are generally effective at capturing other household pests, such as cockroaches, ants, and spiders [26,27,28,29,30]. Despite the documented ineffectiveness of sticky traps [8] and their exclusion from various reviews of bed bug monitoring devices [2,5,31], many pest control companies continue to use them. Surveys reveal that 44% of pest management firms in the United States and 18.5% in Australia still employ sticky traps for bed bug detection [8]. While pitfall traps are effective at catching bed bugs, they are more expensive than sticky traps and will also require regular maintenance or rotation to prevent the accumulation of dust and debris and the associated potential decreases in efficacy [8,19]. Additionally, pitfall traps may be conspicuous to lodging guests, posing a potential disadvantage in consumer perception of the hospitality industry [4,32]. Given the pest status of bed bugs and the limitations of prevailing monitoring methods, it is necessary to explore alternative and innovative strategies to supplement bed bug detection, monitoring, and control [6].

Insights into such strategies can be gleaned from historical practices. For example, Southeastern European people have traditionally scattered common bean (*Phaseolus vulgaris*) leaves around their beds to trap bed bugs [24,33]. Bean leaves trap bed bugs through a piercing–trapping mechanism (rather than entanglement), wherein the trichomes’ terminal hooks physically pierce the bed bugs’ tarsi [34]. The sites where bed bugs are pierced by these trichomes—specifically the integument between tarsal subsegments and the membrane with microtrichia located on the pretarsal claws—are mechanically weaker than the surrounding parts of the pretarsal claws [35]. In a previous study, adult male bed bugs walking across bean leaves were entrapped quickly and rarely ripped free [34]. Thus, a trap utilizing this piercing–trapping mechanism could be useful to develop a novel tool for bed bug detection, monitoring, and even control.

An experimental monitoring surface derived from bean leaves was recently developed [36] and tested in the current study. This flat, flexible, and discrete leaf-derived trapping material (LDTM) is hypothesized to outperform many commercial traps by effectively trapping bed bugs of all life stages and both sexes. The current study compares the performance of a novel leaf-derived trapping material derived from bean leaves to four selected commercial traps in laboratory experiments: three types of sticky traps and a pitfall trap.

## 2. Materials and Methods

### 2.1. Bed Bugs

The bed bugs used in all experiments were obtained from laboratory colonies of the “Earl” strain, purchased from Sierra Research Laboratories (Modesto, CA, USA). With the use of an artificial feeding system, all bed bugs were provided with defibrinated rabbit blood (HemoStat Laboratories, Dixon, CA, USA). Colonies were maintained at 24–26 °C and 15–30% RH, with a photoperiod of 12:12 (L:D) hr. The bed bugs used in the behavioral assay were prepared by feeding a group of bed bugs until they reached engorgement and keeping them in the laboratory for an additional 7–10 days without a blood meal.

### 2.2. Experimental Arenas

The experimental arenas were 51 cm × 51 cm × 15.25 cm (length × width × height) plastic tubs (Fisherbrand™ Heavy Duty Polypropylene Lab Trays, catalog #: S37146, Thermo Fisher Scientific Inc., Waltham, MA, USA) (Figure 1). The inner and outer surfaces of these arenas were coated with Fluon (Teflon PTFE DISP 30–500 mL, Fuel Cell Earth LLC, Stoneham, MA, USA) to prevent bed bug escape and the entry of other crawling insects. To provide a surface on which bed bugs could crawl, a 50 cm × 50 cm square piece of brown kraft paper (50 lb. Kraft Paper Roll, 36″ × 720′, model #: S-1312, ULINE, Pleasant Prairie, WI, USA) was taped down to the arena bottom using a masking tape (3M 2600 Masking Tape, 1″ × 60 yds, model #: S-14886, ULINE, Pleasant Prairie, WI, USA). After each observational period described below, the brown kraft paper was removed and discarded to eliminate chemical residues, live bed bugs, and eggs. The arena was cleaned with 80% ethanol and allowed time to dry before a new sheet of brown kraft paper was affixed. Four small “awnings” made of square pieces of cardboard (10 cm × 10 cm) were attached to the top corners of the arenas to provide shaded areas, simulating lighting conditions where bed bug monitors would be typically deployed (e.g., under beds, sofas, other furniture items).

### 2.3. Commercial Bed Bug Traps

The commercial trap products used in this study were the following: (1) a flat sticky surface (Harris^®^ Bed Bug Detection Traps, modified by removing the manufacturer’s pre-folded enclosure sides), (2) a sticky partial enclosure (Catchmaster Gluee Louee, #150MBGL, Catchmaster AP&G, Bayonne, NJ, USA, folded as per the manufacturer’s direction), (3) a sticky enclosure or “tent” (Harris^®^ Bed Bug Detection Traps, P.F. Harris Manufacturing Company LLC, Cartersville, GA, USA, unmodified), and (4) a pitfall trap (SenSci Volcano Bed Bug Detector, SenSci, Hamilton, NJ, USA, used without the manufacturer’s associated “lure”) (Table 1). All traps selected in this study had a square or rectangular shape to better fit into the corners of the arena. The commercial traps selected in this study had a similar footprint area to facilitate comparisons. The flat sticky surface was modified from its original configuration (i.e., folds removed) to directly compare to our flat LDTM surface (see below).

### 2.4. Experimental Monitoring Surface

In drawing inspiration from the entrapment mechanism described in Szyndler et al. (2013) [34], a novel leaf-derived trapping material (LDTM) was developed by chemically treating bean leaves (patent pending) [36]. The LDTM was produced in small batches in the laboratory.

The LDTM was affixed to a flat piece of printer paper using liquid glue (Elmer’s Glue-All, Westerville, OH, USA). In all direct comparisons with the respective commercial trap, the LDTM remained flat, except when compared to the “sticky partial enclosure”. In the latter case, the LDTM was placed directly onto the sticky surface of the trap and referred to as the “LDTM partial enclosure”. For comparisons with the sticky enclosure and the pitfall trap, the dimension of the LDTM was exactly matched with that of the footprint area (i.e., the area viewed from a top-down perspective) of the commercial trap (Table 1). For comparisons with the flat sticky surface and the sticky partial enclosure, the dimension of the LDTM was exactly matched with that of the sticky surface area of the commercial trap (Table 1).

### 2.5. Experimental Design

Experiments aimed to compare the capture efficacy of the LDTM to the commercial traps in a series of simultaneous evaluations. Each arena was provisioned with one pair of LDTM surfaces and one pair of the same commercial trap products listed above. Pairs of the same type of traps were placed in opposite corners of the arena (Figure 1). All LDTM surfaces and commercial traps were taped down to the brown kraft paper to prevent bed bugs from crawling underneath. Bed bugs were then released at the center of the arena onto a harborage matrix (henceforth referred to as “harborage”), which consisted of a corrugated, clean half filter paper (Whatman Qualitative filter paper, Cat. No. 1001 125, Sigma-Aldrich, Merck KGaA, Darmstadt, Germany) with a small rectangle (2.5 cm × 7.6 cm) of black cardstock affixed over the filter paper to provide a dark environment. Between replications, the placement of each trap pairing was rotated 90° to prevent directional bias.

Bed bugs were released at the center of the experimental arena onto the filter paper of the harborage site, with black cardstock placed over the filter paper to create a dark environment. All trials included a complete scotophase. Each cohort of bed bugs was composed of five adult females, five adult males, five large nymphs (4th–5th instar), and five small nymphs (1st–3rd instar). After 24 h, the total number of bed bugs trapped by each trap was recorded. For this study, the binary distinction between “trapped” and “arrested” will be referred to as the “trap status”. For the flat sticky surface, the sticky partial enclosure, and the sticky enclosure, the bed bugs were considered “trapped” if their body segments or limbs made sufficient contact with the sticky surface, preventing escape. For the pitfall trap, the bed bugs were considered “trapped” if they were entirely inside the reservoir of the pitfall. For the LDTM, bed bugs were considered “trapped” if they were firmly affixed to the material and unable to escape. Unique for the LDTM, some bed bugs were simply arrested on the LDTM or had torn free upon inspection. These bed bugs were recorded as “arrested”. The number of bed bugs on the brown kraft paper but not on traps or in the harborage and bed bugs in the harborage were recorded. The life stage and sex of each bed bug captured were recorded.

All experiments were conducted under natural light conditions, with arenas illuminated solely by an east-facing window; no artificial lights were permitted in the room. For each comparison, the experiments were replicated 12 times.

### 2.6. Statistical Analysis

All statistical analyses were performed using RStudio (Version 2023.06.0 + 421). The data for these analyses included the total number and percentage of bed bugs trapped for each trap type. The data were paired within the replications for the Wilcoxon signed-rank exact tests [37], with 12 replications used in total. In each replication, the total number of bed bugs trapped with the LDTM was compared to the total number trapped with the respective commercial trap. Pooling (from 2 traps of the same type) was conducted to increase the statistical power of the analysis by considering the overall effectiveness of each trap type across the different replications. This approach is justified as the experimental conditions were consistent across replications [38].

To evaluate the effectiveness of different trap types in capturing all developmental stages of bed bugs (i.e., adults and nymphs), trap percentage data were calculated and compared between different trap types. All percentages reported were calculated based on the number of bed bugs trapped and the total number of bed bugs of that respective life stage used for the trial (10 adults and 10 nymphs). Because each assay compared four groups of trapped bed bugs (adults or nymphs caught in either the LDTM or commercial trap), a Friedman test [39] was conducted when comparing the percentage of bed bugs trapped, which were further subdivided by trap type (LDTM vs. a selected commercial trap) and life stage (adult or nymph). The null hypothesis for a Friedman test states that there are no differences among the groups being compared. In other words, it assumes that the dependent variable’s median values (measured on an ordinal scale) are equal across all groups. Conversely, the alternative hypothesis, therefore, suggests that at least two of the groups differ significantly from each other. The post hoc analysis following the Friedman test consisted of Wilcoxon signed-rank tests with Bonferroni correction for multiple comparisons to identify specific differences between groups [38].

To evaluate whether bed bug life stage and trap status (i.e., “trapped” or “arrested”) were independent, the total counts of trapped and arrested bed bugs were summed by life stage (adults and nymphs), and one of two tests of independence were used: Pearson’s chi-squared test [40] or Fisher’s exact test [41] when cell counts were less than 5. For both tests, independence between life stage and trap status was indicated by *p* > 0.05, while a significant association between life stage and trap status would be indicated by *p* < 0.05.

## 3. Results

### 3.1. LDTM Trap Performance Compared to Selected Commercial Traps

In simultaneous comparisons, the LDTM trapped significantly more bed bugs than three of the four commercial traps tested: LDTM vs. flat sticky surface (Wilcoxon signed-rank exact test, *V* = 0, *p* = 0.002), LDTM vs. sticky enclosure (Wilcoxon signed-rank exact test, *V* = 7.5, *p* = 0.046), and LDTM vs. pitfall trap (Wilcoxon signed-rank exact test, *V* = 1.5, *p* = 0.003). The comparison between the LDTM partial enclosure and the sticky partial enclosure showed no significant difference (Wilcoxon signed-rank exact test, *V* = 24.5, *p* = 0.269) (Figure 2a, Table 2). The total number of bed bugs trapped by each trap type and the number not captured by the LDTM were recorded (Table 2). Additionally, the number of bed bugs found in the harborage and elsewhere in the arena after 24 h was documented, with an average of 56.4% of bed bugs remaining in the harborage across all direct comparisons (Table 2).

### 3.2. Trapping Efficiency of LDTM and Commercial Traps for Adult and Nymphal Bed Bugs

Trap frequency varied significantly between trap types and life stages, with the LDTM capturing a higher percentage of nymphs than adults in three of the four direct comparisons. LDTM vs. flat sticky surface: The percentage of nymphs trapped on the LDTM was significantly higher than the percentage of adults trapped on the LDTM (Friedman *χ*^2^ = 24.3, *df* = 3, *p* < 0.001) (Figure 2b). The percentage of nymphs trapped on the LDTM was significantly higher than the percentage of both adults and nymphs trapped on the flat sticky surface (Friedman *χ*^2^ = 24.3, *df* = 3, *p* < 0.001) (Figure 2b). LDTM vs. sticky enclosure: The percentage of nymphs trapped on the LDTM was significantly higher than the percentage of adults trapped on the LDTM (Friedman *χ*^2^ = 17.2, *df* = 3, *p* < 0.001) (Figure 2b). The percentage of nymphs trapped on the LDTM was significantly higher than the percentage of both adults and nymphs trapped in the sticky enclosure (Friedman *χ*^2^ = 17.2, *df* = 3, *p* < 0.001) (Figure 2b). LDTM vs. pitfall trap: The percentage of nymphs trapped on the LDTM was significantly higher than the percentage of nymphs trapped in the pitfall traps (Friedman *χ*^2^ = 23.0, *df* = 3, *p* < 0.001) (Figure 2b). Within the LDTM, the percentage of trapped nymphs was significantly higher than the percentage of trapped adults (Friedman *χ*^2^ = 23.0, *df* = 3, *p* < 0.001) (Figure 2b). LDTM partial enclosure vs. sticky partial enclosure: There were no significant differences in the percentages of bed bugs trapped in the direct comparisons between the LDTM partial enclosure and the sticky partial enclosure (Friedman *χ*^2^ = 2.39, *df* = 3, *p* = 0.495) (Figure 2b).

### 3.3. Effect of Bed Bug Life Stage on LDTM Trap Status

There was a significant association between life stage and trap status for bed bugs when comparing the LDTM to the flat sticky surface (Pearson’s *χ*^2^ = 16.48, *p* < 0.00005, *df* = 1), the LDTM to the sticky enclosure (Fisher’s exact test, *p* < 0.0005), and the LDTM to the pitfall trap (Pearson’s *χ*^2^ = 7.39, *p* < 0.01, *df* = 1). However, there was no significant association between life stage and trapped or arrested bed bugs when comparing the LDTM to the sticky partial enclosure (Fisher’s exact test, *p* = 0.11).

## 4. Discussion

This study demonstrates that the experimental LDTM trapped more bed bugs than the flat sticky surface, the sticky enclosure, and the pitfall trap (Table 2). When directly compared to the selected commercial traps, the LDTM is 40 times more effective than the flat sticky surface (40 vs. 1 for the LDTM and the flat sticky surface, respectively), 2.5 times more effective than the sticky enclosure (52 vs. 21 for the LDTM and sticky enclosure, respectively), and 6 times more effective than the pitfall trap (34 for the LDTM vs. 6 for the pitfall trap, respectively) (Table 2). The last comparison is noteworthy, as pitfall traps are widely regarded as some of the most effective devices for bed bug detection, monitoring, and trapping [5,6,20,22]. However, it is important to note that while previous studies have tested pitfall traps over durations ranging from one to ten weeks [20,22], the comparisons in this study were conducted over 24 h periods.

The increase in trap percentage for the LDTM when compared to the flat sticky surface, the sticky enclosure, and the pitfall trap was primarily driven by the higher efficiency of the LDTM in capturing nymphal bed bugs (Figure 2b). Nymphs are more abundant than adult bed bugs in typical infestations [15], with one unpublished study noting that nymphs comprised 80% of the field population [12]. Physically trapping nymphal bed bugs permanently would prevent them from reaching adulthood, potentially serving as an effective method for controlling field bed bug populations. Therefore, these findings suggest that the LDTM may be an effective tool for monitoring field bed bug populations at all life stages.

The LDTM partial enclosure trapped a comparable number of bed bugs compared to the sticky partial enclosure (Table 2). However, the LDTM partial enclosure trapped fewer nymphs compared to the LDTM in its other direct comparisons in this study. The differences in the LDTM configuration between the two forms of LDTM (the default, flat LDTM, and the LDTM partial enclosure) may partially account for the variation in performance. For the LDTM partial enclosure, the LDTM was placed directly on the sticky surface of the commercial partial enclosure, resulting in a larger thickness of the trapping surfaces (1.03 mm for the LDTM partial enclosure vs. 0.01 mm for the flat LDTM). Given the smaller size of bed bug nymphs (1–3 mm) [1,18], this increased height may have prevented them from effectively reaching the LDTM surface of the LDTM partial enclosure, leading to a reduced number of nymphs trapped. It remains unclear whether the sticky surface of the commercially available sticky trap negatively interfered with the LDTM.

The difference in trapping efficacy between adult and nymphal bed bugs on the LDTM (Figure 2b) may be due to the larger size of adults, which enables them to generate stronger forces, allowing for more frequent escapes compared to nymphs. Previous measurements of bed bugs walking on smooth resin found that adult male bed bugs generated tibial attachment forces twice as large as adult females, and females generated forces 10 times greater than nymphs [42]. The larger force generation of adults relative to nymphs may explain the observed life stage differences in LDTM trap status (i.e., “trapped” vs. “arrested”) (see Section 3.3).

A previous study found that bed bugs walking on kidney bean leaves were trapped after completing a median of nine locomotory cycles (range: 0–39), where one “locomotory cycle” is defined as a single step taken by each of the six legs [34]. In the present study, some bed bugs interacted with the LDTM but were “arrested” (Table 2), possibly because they did not complete the necessary number of locomotory cycles required for capture. Alternatively, some of these bed bugs may have initially been trapped but escaped before full capture, as observed in rare occurrences by Szyndler et al. (2013) [34]. It is also possible that bed bugs not found on the traps after 24 h had escaped the LDTM entirely.

Bed bugs interacted with the LDTM more than the commercial traps at the end of the 24 h trials (i.e., the summation of the “Number of bed bugs trapped by LDTM” and “Number of bed bugs ‘arrested’ by LDTM” bed bugs in Table 2). Bed bugs interacted with the LDTM 71 times more than the flat sticky surface (71 vs. 1, for the LDTM and the flat sticky surface, respectively), 2.6 times more than the sticky enclosure (56 vs. 21, for the LDTM and the sticky enclosure, respectively), and 10.3 times more than the pitfall trap (62 vs. 6, for the LDTM and the pitfall trap, respectively) (Table 2). Bed bugs interacted with the LDTM partial enclosure 2.6 times more than the sticky partial enclosure (44 vs. 17, for the LDTM partial enclosure and the sticky partial enclosure, respectively) (Table 2). The non-smooth texture of the LDTM may provide unique tactile cues that increase the amount of time bed bugs spend on the material. The LDTM utilizes the same piercing–trapping mechanism previously described [34,43], where bed bugs risk self-impalement by the trichomes as they walk across the surface. This self-impaling trapping mechanism significantly reduces the insect’s ability to escape; ripping free from fresh bean leaves is rare and often results in another self-impalement [34]. While this may seem like suboptimal trap performance, the interaction of bed bugs with the LDTM is beneficial as it increases the likelihood of eventual capture. Future research should evaluate the performance of the LDTM over extended periods (e.g., ≥1 week) to fully assess its trapping efficacy and longevity. A future study will assess the number of locomotory cycles necessary for bed bugs of different life stages to be trapped by the LDTM.

The LDTM may be engineered into various configurations and placed strategically within living spaces, such as under furniture legs, wrapped around furniture, or adhered to mattress corners, serving as barriers for bed bugs attempting to reach a host. It could also function as a central component of a harborage trap, attracting bed bugs and physically removing them from the population. For the LDTM to be viable in integrated pest management (IPM) programs, it must be inexpensive, discreet, and commercially available, with a long shelf life and consumer-friendly packaging. While the present study suggests strong viability of the LDTM as a bed bug monitor, future studies should assess its effectiveness in long-term surveys (e.g., ≥1 week) and its potential as a control method for larger bed bug populations.

## Figures and Tables

**Figure 1 insects-16-00362-f001:**
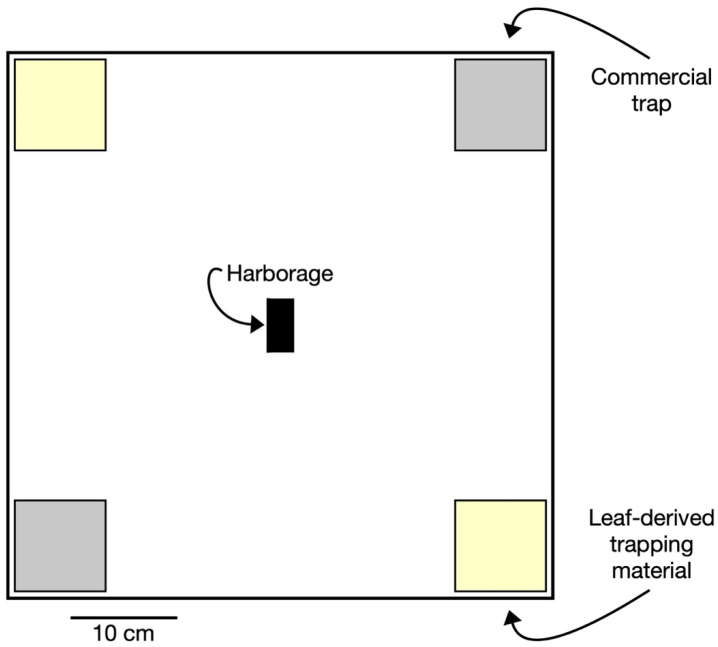
Arena for comparing the leaf-derived trapping material to a commercial trap. Corners diagonal from each other were the same trap type. The bed bugs were released at the harborage site in the middle of the arena, portrayed by the black rectangle.

**Figure 2 insects-16-00362-f002:**
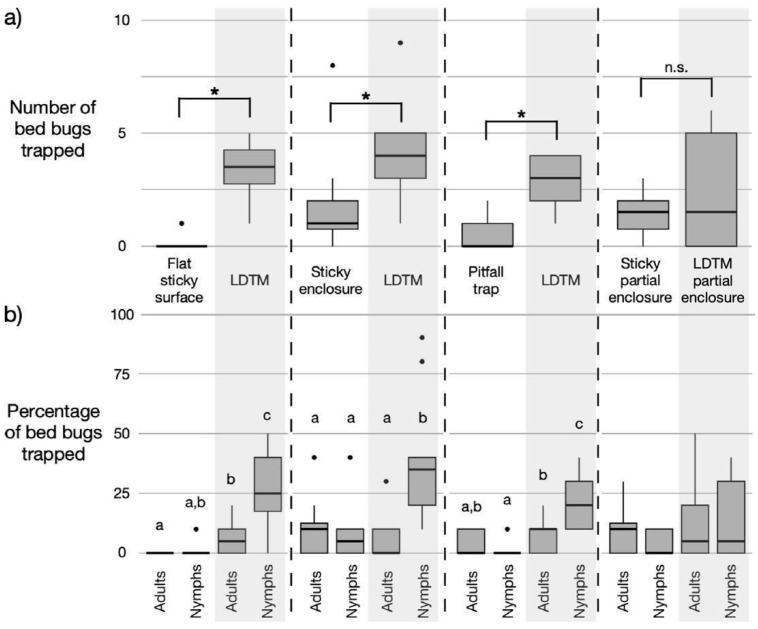
(**a**) The total number of bed bugs trapped after 24 h with different trap types (*n* = 12 trials; 240 bed bugs: 120 adults, 120 nymphs). Vertical dashed lines delineate direct comparisons between the leaf-derived trapping material (LDTM) and each commercial trap: a flat sticky surface (Harris Bed Bug Detection Trap with folds removed), a sticky enclosure (Harris Bed Bug Detection Trap), a pitfall trap (Volcano), and a sticky partial enclosure (Catchmaster Gluee Louee). Asterisks indicate significant differences between paired assays (*p* ≤ 0.05, Wilcoxon signed-rank test), while “n.s.”. denotes no statistically significant difference (*p* > 0.05). (**b**) The percentage of bed bugs trapped across different trap types for adults and nymphs. The horizontal line within each box represents the median. The lower and upper bounds of the box indicate the first and third quartiles, respectively. The lines (“whiskers”) extending from the boxes correspond to the minimum and maximum values. Outliers are identified as dots. Box plots followed by different letters are significantly different based on the Wilcoxon signed-rank test with adjustments for multiple comparisons (Bonferroni correction).

**Table 1 insects-16-00362-t001:** The selected commercial traps with their product name, their dimensions, and a description of their characteristics.

Trap Design	Commercial Trap Product Name	Commercially Available Trap Product Image	Dimensions	Description
Flat sticky surface	Harris Bed Bug Detection Trap with folds removed	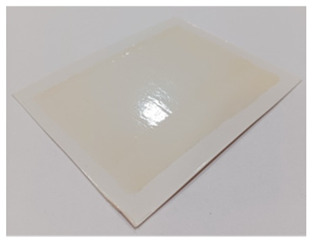	Sticky surface area: 6.7 cm × 8.5 cm	This is a modified version of the sticky tent (see the row below) with the folds removed, leaving only the sticky surface exposed.
Sticky enclosure	Harris Bed Bug Detection Trap	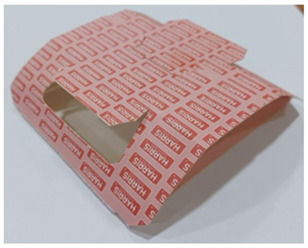	Trap footprint area: 10.0 cm × 8.1 cm	All four sides of the trap fold to form a tent shape, each featuring an opening for bed bugs to enter. The waxy inner surface prevents bed bugs from escaping, and the base of the trap has a sticky surface.
Pitfall trap	Volcano	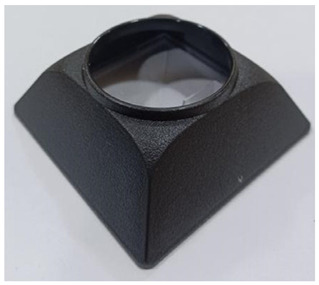	Trap footprint area: 7.2 cm × 7.2 cm	The trap has a pyramidal shape with a slanted, bumpy exterior that allows bed bugs to climb up. The smooth inner surface prevents escape. This study did not use the chemical lure module.
Sticky partial enclosure	Catchmaster Gluee Louee	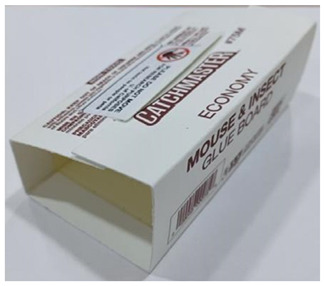	Sticky surface area: 13.5 cm × 7.0 cm	Two sides of the trap fold, while the remaining two sides remain open to allow insects to crawl into the trap. The interior has a sticky surface.

**Table 2 insects-16-00362-t002:** The total number of bed bugs captured after 24 h with the leaf-derived trapping material (LDTM) and selected commercial traps when compared directly across all replicates (*n* = 12 simultaneous comparison assays, 240 bed bugs). The table also includes counts of bed bugs observed in the harborage matrix, elsewhere in the arena, and those classified as “arrested” by the LDTM after 24 h. Bed bugs were deemed “arrested” if they were not fully trapped on the LDTM or had torn free upon inspection.

Direct Comparison	Number of Bed Bugs Trapped with Commercial Traps	Number of Bed Bugs Trapped with LDTM	Number of Bed Bugs in Harborage	Number of Bed Bugs Elsewhere in the Arena	Number of Bed Bugs “Arrested” with LDTM
Flat sticky surface vs. LDTM	1	40	141	27	31
Sticky enclosure vs. LDTM	21	52	113	50	4
Pitfall trapvs. LDTM	6	34	145	27	28
Sticky partial enclosure vs. LDTM partial enclosure	17	29	142	37	15

## Data Availability

The raw data supporting the conclusions of this article will be made available by the authors on request.

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
