# Peer review of "A Novel Leaf-Derived Trapping Material Is More Effective at Capturing Common Bed Bugs (Hemiptera: Cimicidae) than Selected Commercial Monitoring Devices"

_insects, 2025, doi:10.3390/insects16040362_

Round 1
Reviewer 1 Report
Comments and Suggestions for Authors
This study compared a novel man-made material (LDTM) with different sticky traps and SenSci volcano bed bug monitor for trapping bed bugs. The subject is very interesting to readers. However, this study has two major defects:
First, the title is scientifically wrong. The results are based on comparisons with ineffective commercial products for bed bug monitoring. Sticky traps are known not effective for detecting bed bugs. They are not designed as bed bug monitors. Many pitfall-style bed bug monitors are available. But only a few are effective. The authors selected SenSci monitor, which is not an effective monitor and is much less effective than Climbup interceptors and a few other monitors with soft exterior walls. In our study, Climbup HD black is 14x more effective than SenSci volcano bed bug monitor. The current experimental design only provides very limited and biased information about LDTM as a bed bug monitoring material. Authors should conduct additional experiments comparing LDTM with “Climbup HD Black” or another monitor with soft exterior walls. They would have different conclusions on the relative efficacy of LDTM and pitfall-style monitors.
Second, Figure 2 shows that each trap trapped an average 1-4 bed bugs. I am highly suspicious that these small numbers are statistically valid for comparing different traps. Replicates where the total number of trapped bed bugs in an arena is too small (such as < 15) should be excluded from analysis. Authors either need to add more replicates or release much larger numbers of bed bugs per arena. The small number of trapped bed bugs makes it especially difficult to compare the bed bug stages for each type of trap in Table 3.
Author Response
We want to thank the reviewer for providing their feedback. The peer-review process is essential for improving our manuscript, and we appreciate the reviewers' service. Please find our point-by-point responses below.
Comment 1: First, the title is scientifically wrong. The results are based on comparisons with ineffective commercial products for bed bug monitoring. Sticky traps are known not effective for detecting bed bugs. They are not designed as bed bug monitors. Many pitfall-style bed bug monitors are available. But only a few are effective. The authors selected SenSci monitor, which is not an effective monitor and is much less effective than Climbup interceptors and a few other monitors with soft exterior walls. In our study, Climbup HD black is 14x more effective than SenSci volcano bed bug monitor. The current experimental design only provides very limited and biased information about LDTM as a bed bug monitoring material. Authors should conduct additional experiments comparing LDTM with “Climbup HD Black” or another monitor with soft exterior walls. They would have different conclusions on the relative efficacy of LDTM and pitfall-style monitors.
Response to the first paragraph:
- We have made the appropriate changes to the title.
- We agree with the reviewer that sticky traps are ineffective bed bug monitors. However, many pest management professionals still use sticky traps in the field. Furthermore, numerous sticky traps are advertised as bed bug traps. The Introduction addressed this statement (see Lines 57-65).
- We are aware that there are multiple types of pitfall traps. However, we felt that the pitfall trap we selected would suffice. We selected the SenSci pitfall trap for two reasons: 1) the pitfall trap has a square geometry, so it fit more effectively in the corners of the square arena used in this study, and 2) the SenSci pitfall trap had an area comparable to the other traps (i.e., the selected glue traps) selected in the present study (i.e,. the sticky traps). It was important to keep the size and shape consistent between the LDTM and the commercial traps that were being compared. This facilitates direct comparisons between the different trap types used in the study. To describe this clarification, we have added the appropriate language to Lines 126-128 (“All traps selected in this study had a square or rectangular shape to better fit into the corners of the arena. The commercial traps selected in this study had a similar footprint area to facilitate comparisons.”)
Comment 2: Second, Figure 2 shows that each trap trapped an average 1-4 bed bugs. I am highly suspicious that these small numbers are statistically valid for comparing different traps. Replicates where the total number of trapped bed bugs in an arena is too small (such as < 15) should be excluded from analysis. Authors either need to add more replicates or release much larger numbers of bed bugs per arena. The small number of trapped bed bugs makes it especially difficult to compare the bed bug stages for each type of trap in Table 3.
Response to the second item:
While the number of bed bugs trapped varied, the statistical tests support our conclusions. The current study used 20 bed bugs per replication to simulate a low-level infestation. The arena selected for this study was also not incredibly large (51 cm x 51 cm). To eliminate redundancies, we have removed Table 3 from the manuscript. All text within the manuscript has reflected this change appropriately.
Reviewer 2 Report
Comments and Suggestions for Authors
The manuscript: A novel leaf-derived trapping material is more effective at capturing bed bugs (Hemiptera: Cimicidae) than commercially available monitoring devices provides robust data comparing leaf-derived trapping material (LDTM) to existing bed bug trapping technologies. The authors are well aware of the strengths limitations of this study and discuss further work to be done coherently The manuscript is well written and does not have any significant issues with clarity of English expression. The paper will provide a significant data point for possibly developing these papers for trapping. There are generally significant effects of the paper that support the line of research the team is undertaking. The following minor comments are provided.
Ln 177-178 I am confused by the possibility of being on the paper, but not the trap. Aren’t all the papers in traps? Or they really are traps themselves. This section needs to be rewritten for clarity.
Figure 2: what is represented in the box plots? Confidence intervals. What are the horizontal lines, means? Medians?
Table 3 provides redundant information versus what was presented in Fig 2 and Table 2.
Is it possible to elaborate further upon the expected shelf life of this paper? Beyond longer duration trials, It would also be interesting to conduct some trials where the paper is different ages before even being used.
Would it be possible to manufacture a duokicated version of the paper using nanofabrication techniques?
Author Response
We want to thank the reviewer for providing their feedback. The peer-review process is essential for improving our manuscript, and we appreciate the reviewers' service. Please find our point-by-point responses below.
Comment 1: Ln 177-178 I am confused by the possibility of being on the paper, but not the trap. Aren’t all the papers in traps? Or they really are traps themselves. This section needs to be rewritten for clarity.
Response to the first item (regarding Lines 177-178):
Thank you for asking this question and paying close attention to our methods. It was after this question that we noticed the methods required clarification. This study uses various types of paper, and the novel material in this study is not a paper. The leaf-derived trapping material (LDTM) is glued onto a piece of printer paper. This printer paper is taped onto a brown kraft paper (disposable) to easily replace this brown kraft paper between trials and clean the arena. The different paper types required clarification, and we have changed the text describing this process (i.e., “Experimental arenas” Lines 103-117, 155) to reflect this clarification.
Comment 2: Figure 2: what is represented in the box plots? Confidence intervals. What are the horizontal lines, means? Medians?
Response to the second item (regarding Figure 2):
Thank you for this question. This is also an item we did not initially include in our manuscript. The horizontal line represents the median. The box below the median represents the first quartile. The box above the median represents the third quartile. The lines protruding from the box plots begin and end at the minimum and maximum, respectively. This is further clarified in Figure 2’s caption (“The horizontal line within each box represents the median. The lower and upper bounds of the box indicate the first and third quartiles, respectively. The lines (“whiskers”) extending from the boxes correspond to the minimum and maximum values. Outliers were identified as dots.”)
Comment 3: Table 3 provides redundant information versus what was presented in Fig 2 and Table 2.
Response to the third item (regarding Table 3): We have removed Table 3 from the manuscript. All text within the manuscript has reflected this change appropriately.
Comment 4: Is it possible to elaborate further upon the expected shelf life of this paper? Beyond longer duration trials, It would also be interesting to conduct some trials where the paper is different ages before even being used.
Response to the fourth item (expected shelf life of the material): The material is relatively new and thus is being tested as time progresses. Unpublished observations show no difference in trapping efficacy between a newly manufactured material and one manufactured 12 months beforehand. The current scope of the manuscript does not test the different ages of the material, but future studies can address this question.
Comment 5: Would it be possible to manufacture a duokicated version of the paper using nanofabrication techniques?
Response to the fifth item (manufacturing a duplicated version): One of the co-authors is an inventor on patents for using microfabrication approaches, which are being actively pursued in the laboratory.
Reviewer 3 Report
Comments and Suggestions for Authors
This study describes a new type of bug monitor/trap and compares it with existing bug monitoring products. This study is very interesting and eye-catching. However, there are still some noteworthy shortcomings.
1. Authors only selected one type of pitfall trap, in fact, there are some other kinds of pitfall traps commonly used in USA, such as ClimbUp. Is the LDTM trap created by authors more effective than ClimbUp?
2. In this study, only the common bed bugs were selected, thus, I suggest the authors indicate "the common bed bugs" in the article title.
3. In this paper, bed bugs were not adapted to the environment within the experimental area, but were directly placed onto a harborage, which differs from the actual situation. Just placed in the experimental area, bed bugs are likely to run around in panic due to disturbance, resulting in a higher number of bed bugs captured by the monitor. And for bedbugs that have fully adapted to the environment, can the monitor still achieve ideal results?
Author Response
Reviewer 3
We want to thank the reviewer for providing their feedback. The peer-review process is essential for improving our manuscript, and we appreciate the reviewers' service. Please find our point-by-point responses below.
Comment 1: Authors only selected one type of pitfall trap, in fact, there are some other kinds of pitfall traps commonly used in USA, such as ClimbUp. Is the LDTM trap created by authors more effective than ClimbUp?
Response to 1: We are aware that there are multiple types of pitfall traps. However, we felt that the pitfall trap we selected would suffice. We selected the SenSci pitfall trap for two reasons: 1) the pitfall trap has a square geometry, so it fit more effectively in the corners of the square arena used in this study, and 2) the SenSci pitfall trap had an area comparable to the other traps (i.e., the selected glue traps) selected in the present study (i.e,. the sticky traps). It was important to keep the size and shape consistent between the LDTM and the commercial traps that were being compared. This facilitates direct comparisons between the different trap types used in the study. To describe this clarification, we have added the appropriate language to Lines 126-128 (“All traps selected in this study had a square or rectangular shape to better fit into the corners of the arena. The commercial traps selected in this study had a similar footprint area to facilitate comparisons.”)
Comment 2: In this study, only the common bed bugs were selected, thus, I suggest the authors indicate "the common bed bugs" in the article title.
Response to 2: Thank you for this suggestion. We have incorporated this change into the title.
Comment 3: 3. In this paper, bed bugs were not adapted to the environment within the experimental area, but were directly placed onto a harborage, which differs from the actual situation. Just placed in the experimental area, bed bugs are likely to run around in panic due to disturbance, resulting in a higher number of bed bugs captured by the monitor. And for bedbugs that have fully adapted to the environment, can the monitor still achieve ideal results?
Response to 3:
While we agree with the sentiment in this comment, all comparisons were fair and equal—that is, all bugs were allowed to wander from the same harborage site at the same time. Bed bugs had the opportunity to leave the harborage or stay within the harborage. Indeed, in column 4 of Table 2, most of the bed bugs were still found within the harborage site after 24 hours for most two-way comparisons. The percentage of bed bugs which were found in the harborage site after 24 hours ranged from 47% - 60% (Flat sticky surface vs. LDTM: 58.8%, Sticky enclosure vs. LDTM: 47.1%, Pitfall trap vs. LDTM, 60.4%, and Sticky partial enclosure vs. LDTM partial enclosure: 59.2%).
The bed bugs were placed on the harborage site's filter paper; then, the black cardstock was placed over the filter paper to provide a localized dark environment for the bugs. This clarification was added to the manuscript (Lines 168-170, “Bed bugs were released at the center of the experimental arena onto the filter paper of the harborage site, with black cardstock placed over the filter paper to create a dark environment”).
A separate field study would be important to compare the LDTM and other commercial bed bug monitors in a more realistic setting. A separate field study, in which bed bugs were already present in an environment before the traps were placed, was recently conducted by the co-authors of this group. The data for that study is currently being prepared for publication.
Round 2
Reviewer 1 Report
Comments and Suggestions for Authors
I think the authors' reply is not satisfactory. The scientific value of this study is very limited.